# The GPR39 Receptor Plays an Important Role in the Pathogenesis of Overactive Bladder and Corticosterone-Induced Depression

**DOI:** 10.3390/ijms252312630

**Published:** 2024-11-25

**Authors:** Jan Wróbel, Paulina Iwaniak, Piotr Dobrowolski, Mirosława Chwil, Ilona Sadok, Tomasz Kluz, Artur Wdowiak, Iwona Bojar, Ewa Poleszak, Marcin Misiek, Łukasz Zapała, Ewa M. Urbańska, Andrzej Wróbel

**Affiliations:** 1Medical Faculty, Medical University of Lublin, 20-093 Lublin, Poland; wrobeljan@onet.eu; 2Department of Experimental and Clinical Pharmacology, Medical University of Lublin, Jaczewskiego 8b, 20-090 Lublin, Poland; ewa.urbanska@umlub.pl; 3Department of Functional Anatomy and Cytobiology, Faculty of Biology and Biotechnology, Maria Curie-Sklodowska University, Akademicka 19, 20-033 Lublin, Poland; 4Department of Botany and Plant Physiology, University of Life Sciences in Lublin, 15 Akademicka St., 20-950 Lublin, Poland; miroslawa.chwil@up.lublin.pl; 5Department of Chemistry, Institute of Biological Sciences, Faculty of Medicine, Collegium Medicum, The John Paul II Catholic University of Lublin, Konstantynów 1J, 20-708 Lublin, Poland; ilona.sadok@kul.pl; 6Department of Gynecology, Gynecology Oncology and Obstetrics, Institute of Medical Sciences, Medical College of Rzeszow University, Rejtana 16c, 35-959 Rzeszow, Poland; jtkluz@interia.pl; 7Obstetrics and Gynecology, Faculty of Health Sciences, Medical University of Lublin, 4-6 Staszica St., 20-081 Lublin, Poland; wdowiakartur@gmail.com; 8Department of Women’s Health, Institute of Rural Health in Lublin, Jaczewskiego 2 St., 20-090 Lublin, Poland; iwonabojar75@gmail.com; 9Laboratory of Preclinical Testing, Department of Applied and Social Pharmacy, Medical University of Lublin, 1 Chodźki St., 20-093 Lublin, Poland; ewapoleszak@umlub.pl; 10Department of Gynecologic Oncology, Holy Cross Cancer Center, 25-377 Kielce, Poland; marcin.misiek@onkol.kielce.pl; 11Clinic of General, Oncological and Functional Urology, Medical University of Warsaw, Lindleya 4, 02-005 Warsaw, Poland; lzapala@wum.edu.pl; 12Second Department of Gynecology, Medical University of Lublin, Jaczewskiego 8, 20-090 Lublin, Poland; wrobelandrzej@yahoo.com

**Keywords:** corticosterone, GPR39 agonist, depression, overactive bladder, rats

## Abstract

Despite the close and clinically confirmed association between depression and overactive bladder, it remains unclear whether this affective disorder is a factor causing overactive bladder or whether overactive bladder is a specific symptom of psychosomatic disorders. This study examined the effects of repeated corticosterone administration on the occurrence of symptoms associated with depression and overactive bladder. Additionally, we examined whether administering TC-G 1008, an antidepressant that selectively activates the GPR39 receptor, could alleviate corticosterone-induced depression-like behavior and detrusor overactivity-related changes in cystometric measurements. We also explored its potential to reverse alterations in various biomarkers associated with both conditions in the serum, urinary bladder, and brain of female rats. The administration of corticosterone (20 mg/kg/day for 14 days) yielded anticipated results, including an increase in the duration of immobility during the forced swim test, alterations in parameters specific to bladder overactivity, a decrease in neurotrophins, and an elevation in pro-inflammatory cytokine levels. Treatment with TC-G 1008 (15 mg/kg/day) alleviated symptoms of both detrusor overactivity and depression, while also restoring the levels of biochemical and cystometric markers to normal ranges. Additionally, antidepressants based on GPR39 agonists could enhance the levels of kynurenic acid in the neuroprotective pathway. These results indicate that the GPR39 agonist receptor might be a promising future therapeutic approach for treating overactive bladder that occurs alongside depression.

## 1. Introduction

Affecting individuals across all age groups, overactive bladder (OAB) syndrome is a highly prevalent urologic condition that can significantly diminish one’s quality of life [1,2]. OAB is a condition characterized by symptoms that arise from various underlying causes. These include disorders of the central and peripheral nervous systems, as well as imbalances in local neurotransmitters [3]. Potential anatomical origins of OAB symptoms have been proposed to include the urothelium, suburothelium, detrusor, and urethra [4]. It has been suggested that psychiatric co-occurring conditions may play a part in the experience of urinary urgency. A range of psychiatric disorders are more common among patients suffering from OAB, with depression observed the most frequently [5].

The underlying mechanisms of depression are highly intricate and not yet fully elucidated. Although numerous theories have been proposed over the years, including monoaminergic, neurotrophic, inflammatory, and glutamatergic hypotheses, there is still no unified understanding of how the disorder develops. The lack of consensus regarding depression’s pathogenesis persists, despite the various explanations that have been put forward throughout the decades [6,7,8]. To identify a rapid-onset antidepressant, it is essential to comprehensively elucidate the pathophysiology of the disorder. Excessive glutamate release was implicated as an important factor leading to behavioral abnormalities [9]. Suppressing the glutamatergic system may reestablish equilibrium between glutamate and GABA neurotransmission, potentially inducing an antidepressant effect that surpasses the efficacy of traditional antidepressant medications [10]. As a crucial trace element, zinc influences enzymatic processes, plays a role in immune system function, and serves as an allosteric modulator for the ionotropic N-methyl-D-aspartate (NMDA) receptor [11,12]. Studies have shown a connection between zinc deficiency and clinical major depression, as well as depression-like behaviors in animal experiments. On that basis, a lower level of zinc has been proposed as a marker of depression [13,14,15]. Apart from NMDA, zinc may impact the activity of other ionotropic glutamate receptors [11] and was shown to exert an antidepressant-like properties in both preclinical and clinical studies. Significantly, zinc plays a crucial role in upregulating both BDNF mRNA and protein expression. Conversely, a lack of zinc in the body leads to decreased levels of neurotrophic factors within the brain.

Additionally, zinc functions as a natural agonist for the G protein-coupled receptor (GPR39) receptor, potentially playing a significant role in depression’s pathophysiology [16,17]. The GPR39 receptor is found in various organs, including the pancreas, kidney, liver, and gastrointestinal tract. This receptor is classified as a member of the class A rhodopsin receptors [18]. GPR39 is also expressed in cortical, hippocampal, and amygdala neurons, which play a role in emotional processing [18,19]. GPR39 activates primarily Gq and G12/13 signaling pathways. Additionally, it can be further activated by zinc ions, which trigger Gs signaling and initiate the ERK1/2 MAPK signaling cascade [16]. New evidence suggests that GPR39 plays a significant role in the development of depression. The reduction in GPR39 levels was detected in the hippocampus and cortex of individuals who died by suicide [20]. Transgenic mice with deleted GPR39 gene showed a depression- and anxiety-like phenotype [17]. Additionally, the absence of GPR39 eliminates the antidepressant effects of SSRIs (selective serotonin reuptake inhibitors), which suggests that the GPR39 receptor may play a role in the mechanism of action of these drugs [21]. The findings suggest that GPR39 could be a potential candidate for developing novel antidepressant medications.

A growing body of data indicates a strong connection between inflammatory processes, stress, and depressive disorders [22,23]. Tryptophan (TRP) metabolism along the kynurenine pathway (KP) yields a number of metabolites known to interact with glutamate receptors and to exert multiple effects on cellular survival, brain development, and immune function [24,25]. Noteworthy, the KP is easily activated during inflammation, which constitutes a broadly accepted feature of depression. Among KP metabolites able to interfere with glutamatergic receptors, NMDA receptor agonist quinolinic acid (QA) and broad-spectrum glutamatergic antagonist with the highest affinity for NMDA receptors, kynurenic acid (KYNA), are the focus of scientific attention [25]. Accumulated evidence indicates that during pro-inflammatory conditions, TRP metabolism is shifted to QA at the expense of KYNA [26,27]. In consequence, excitotoxicity and impaired hippocampal neurogenesis may contribute to the pathogenesis of depression [28].

The connection between OAB and depression is well recognized. A systematic review revealed that a positive correlation between overactive bladder (OAB) and depression was a common observation and occurred in 26 studies; however, 11 reports did not show such correlation [29]. The level of stress increases as a result of severe decline in a patient’s quality of life due to OAB symptoms. This, in turn, may trigger depression and other affective disorders [30]. On the other hand, depression may be a risk factor for a new-onset OAB [31].

However, the precise links between inflammation and KYNA levels on one side, and depression and OAB on the other, remain to be established and require further investigation. The relationship between depression and overactive bladder (OAB) remains unclear. The contribution of existing OAB symptoms to depression or, alternatively, of depression to chronic OAB symptoms is not recognized. Currently, no studies have examined the potential impact of GPR39 receptors on depression that occurs alongside bladder overactivity. We propose that the activation of GPR39 could be a potential approach for treating these co-occurring conditions.

The goal of this study was to determine whether an acute administration of selective agonist of the GPR39 receptor TC-G 1008 would affect OAB, depressive-like symptoms, immune status, and KYNA levels in the corticosterone model of depression in female rats.

## 2. Results

### 2.1. The Influence of TC-G 1008 on Corticosterone Changes in Cystometric Parameters

Consistent with an agent that elicited a response akin to OAB, CORT induced alterations in both the storage- and voiding-phase cystometric parameters (Figure 1). During the storage phase, several parameters showed improvement: bladder pressure (BP), detrusor overactivity index (DOI), threshold pressure (TP), frequency of non-voiding contractions (FNVC), volume threshold for triggering NVC (VTNVC), amplitude of non-voiding contractions (ANVC), and bladder compliance (BC) (Figure 1). During the voiding stage, indicators of symptomatic overactive bladder (OAB) were observed, including increases in the voided volume (VV), intercontraction interval (ICI), and area under the pressure curve (AUC). Notably, rats administered a combination of CORT (20 mg/kg/day) and TC-G 1008 (15 mg/kg/day) exhibited reversed effects. Specifically, the storage-phase parameters FNVC, DOI, and BP showed reductions of 50%, 77%, and 46%, respectively, when compared to the CORT group. Additionally, the combination group exhibited improvements in various voiding- and storage-phase parameters, suggesting a reversal of CORT-induced symptoms. Voiding parameters that increased included VV (39%), ICI (30%), and AUC (27%), while storage-phase parameters showing improvement were TP (35%), VTNVC (39%), and BC (51%). The analysis revealed no significant statistical variations in post-void residual (PVR) and maximum voiding pressure (MVP) measurements, with one exception: the CORT + GPR group exhibited a 21% lower MVP level (*p* = 0.03) when compared to the GPR group (Figure 1).

### 2.2. The Effects of TC-G 1008 on Corticosterone-Induced Behavioral Changes

Results from the forced swim test (FST) demonstrated a significant difference in immobility time. Rats subjected to a 14-day CORT treatment at 20 mg/kg/day exhibited a notable 20% increase in immobility duration compared to the control group (*p* < 0.001). The administration of TC-G 1008 (15 mg/kg/day) effectively counteracted this CORT-induced depressive-like behavior (*p* < 0.001), as shown in Figure 2A. Figure 2B illustrates that the administration of TC-G 1008 (15 mg/kg/day), either alone or combined with CORT, had no significant effect on the animals’ locomotor activity (*p* = 0.85).

### 2.3. Selected Biochemical Analysis

Significantly, the intraperitoneal delivery of TC-G 1008 (15 mg/kg/day) did not produce any observable impact on the function of the urinary bladder or micturition patterns in healthy rats. However, TC-G 1008 reduced the intensity of overactive bladder (OAB) induced by corticosterone (CORT).

#### 2.3.1. Corticotropin-Releasing Factor (CRF) Levels

The CORT treatment (20 mg/kg/day) administered to rats resulted in notably increased CRF levels across all examined samples. Specifically, the hippocampus showed an increase of approximately 246%, the prefrontal cortex exhibited a rise of about 248%, and plasma levels were elevated by 28% compared to the control group. Simultaneous treatment with TC-G 1008 (15 mg/kg/day) reversed the noxious effects of corticosterone in the plasma (27%, *p* < 0.001) and tested brain areas, the hippocampus (138%, *p* < 0.001) and prefrontal cortex (156%, *p* < 0.001) (Figure 3).

#### 2.3.2. CGRP, OCT3, TRPV1, and ATP Levels in the Bladder Urothelium

Rats given the CORT treatment (20 mg/kg/day) had over 3.6 times higher ATP levels compared to the control (*p* < 0.001), and administration of TC-G 1008 (15 mg/kg/day) reversed this effect almost to the control level (Figure 3).

Following a 14-day treatment with CORT (20 mg/kg/day), the examined parameters in the rats’ urothelium showed significant increases. The CGRP level rose by approximately 147% compared to the control group, TRPV1 by about 1140%, and OCT3 by roughly 400%. A subsequent 7-day intravenous administration of TC-G 1008 (15 mg/kg/day) normalized the CGRP and OCT3 values, bringing them back to baseline control levels. However, this GPR39 agonist only partially reduced the elevated TRPV1 concentration, decreasing it to about 191% (*p* < 0.001) compared to CORT administration alone. Nevertheless, this level remained three times higher than that of the control group (*p* = 0.04) (Figure 4).

#### 2.3.3. VAChT and Rho Kinase (ROCK1) Levels in the Bladder Detrusor Muscle

Figure 4 demonstrates that rats given CORT treatment (20 mg/kg/day) exhibited notably elevated levels of Rho kinase (ROCK1) and VAChT in the bladder detrusor muscle. Compared to the vehicle-treated group, these levels increased by approximately 150% and 182%, respectively. Following the administration of TC-G 1008 therapy (15 mg/kg/day), a significant decrease was observed in both ROCK1 and VAChT levels. ROCK1 showed a reduction of 113% (*p* < 0.001), while VAChT decreased by 32% (*p* = 0.03). However, it is noteworthy that VAChT levels did not revert to their initial baseline values (Figure 4).

#### 2.3.4. Brain-Derived Neurotrophic Factor (BDNF) Levels

Rats treated with CORT (20 mg/kg/day) showed markedly reduced BDNF levels in the hippocampus (27% decrease) and prefrontal cortex (24% decrease) when compared to the control group receiving the vehicle. However, BDNF concentrations in urine samples from CORT-treated animals increased substantially, rising by 141%. Treatment with TC-G 1008 reversed the CORT-induced BDNF reductions in the hippocampus (by 50%, *p* < 0.001) and prefrontal cortex (by 18%, *p* = 0.01) but did not affect BDNF levels in plasma. Additionally, TC-G 1008 restored the elevated BDNF levels in the urine. CORT administration led to a substantial increase in urinary BDNF (by 141% vs. controls), which was reduced by 33% (*p* = 0.03) following TC-G 1008 administration. No significant differences in plasma BDNF levels were observed. The protective effects of TC-G 1008 on BDNF levels were significant in the hippocampus (50%, *p* < 0.001) and prefrontal cortex (18%, *p* = 0.01) compared to the CORT group (Figure 4I–L).

#### 2.3.5. Nerve Growth Factor (NGF) Levels

Daily subcutaneous CORT injections (20 mg/kg) led to a notable reduction in NGF concentrations within the examined brain regions. Specifically, the hippocampus experienced a decrease of approximately 41%, while the prefrontal cortex showed a decline of about 46%. Administration of TC-G 1008 at 15 mg/kg/day resulted in a significant restoration of NGF levels in both the hippocampus (49%, *p* < 0.001) and prefrontal cortex (48%, *p* = 0.02). However, the combined treatment of CORT and TC-G 1008 did not significantly affect NGF levels in plasma (Figure 4).

#### 2.3.6. Interleukin 1L-β, IL-6, and IL-10 Levels in Brain Structures and Plasma

The administration of CORT (20 mg/kg/day) resulted in notable increases in IL1-β and IL-6 levels, as anticipated. Specifically, in comparison to the control group, the hippocampus showed elevations of 32% and 45%, respectively, while the prefrontal cortex exhibited increases of 52% and 34%. Additionally, plasma levels rose by 17% for IL1-β and 77% for IL-6. Notably, the plasma concentration of IL-10 remained unchanged (Figure 5). In contrast, TC-G 1008 brought the levels of IL1-β and IL-6 back to normal in the examined brain regions of corticosterone-exposed animals (Figure 5). An analysis revealed a significant decrease in IL1-β CORT-TC-G 1008 interactions for the analysis of the hippocampus (*p* < 0.001) and prefrontal cortex (*p* < 0.001). Finally, our study presented a lower concentration of IL-10 after CORT installation, in certain urogynecological pathologies which cause bladder issues, while the simultaneous TC-G 1008 treatment significantly improved its concentration in the hippocampus (38%, *p* = 0.04), prefrontal cortex (62%, *p* < 0.002), and in plasma (67%, *p* < 0.001) compared to CORT (Figure 5G–I).

#### 2.3.7. Tumor Necrosis Factor α (TNF-α) Levels

Administration of CORT (20 mg/kg/day) to rats resulted in significantly elevated TNF-α levels in both the hippocampus (approximately 19% higher than the vehicle-treated group) and the prefrontal cortex (28% increase). Plasma concentrations of TNF-α remained statistically unchanged (Figure 5). However, administration of TC-G 1008 (15 mg/kg/day) significantly counteracted the detrimental alterations in TNF-α levels caused by corticosterone exposure, particularly in the hippocampus (20%, *p* = 0.009) and prefrontal cortex (22%, *p* = 0.002).

#### 2.3.8. KYNA Levels in Brain Areas and Plasma

Administering glucocorticoid (CORT) at 20 mg/kg/day for two weeks resulted in substantial decreases in KYNA levels across various regions. Specifically, the hippocampus experienced a 43% reduction, the prefrontal cortex showed a 56% decrease, and plasma levels dropped by 60%. TC-G 1008 given simultaneously at a dose of 15 mg/kg/day raised the lowered concentrations of KYNA compared to the basal values for the hippocampus (84%, *p* = 0.002), prefrontal cortex (78%, *p* < 0.001), and plasma (102%, *p* = 0.003), respectively (Figure 6).

### 2.4. Assessing c-Fos Expression Levels in Key Micturition Control Regions

Figure 7 illustrates substantial alterations in c-Fos expression across all examined compartments following CORT administration, when compared to the control group. In contrast, TC-G 1008 exhibited the ability to substantially reduce c-Fos expression following CORT exposure (comparing the CORT-plus-TC-G 1008 group to the CORT group) across all three examined micturition regions: MPA (medial preoptic area) (49%, *p* < 0.001), PMC (pontine micturition center) (60%, *p* < 0.001), and vIPAG (ventrolateral periaqueductal gray) (32%, *p* = 0.03).

## 3. Discussion

The link between OAB and depression has been acknowledged in the literature; yet, the underlying mechanisms are still not understood. The available therapy of OAB is not optimal and novel therapeutic approaches are needed. Conventional treatment includes administration of antimuscarinic agents, agonists of β3-adrenergic receptors, and botulinum toxin A injections into the bladder [32,33,34]. In case of concomitant depression, the therapeutic effect of commonly used drugs develops slowly; furthermore, their adverse effects often limit the clinical use of antidepressants. Thus, a search for new substances that bring relief to people suffering from depression with urinary bladder dysfunction seems essential.

The stimulation of GPR39 receptors was shown to enhance the healing of wounds [35], reduce the severity of inflammatory bowel disease [36], and mitigate symptoms of depression [37]. Furthermore, GPR39 may reduce inflammatory processes and improve both oxidative stress and mitochondrial function [38,39,40]. Increased levels of GPR39 were revealed in the central nervous system during aging [41] and in various types of cancer [42]. GPR39 may alleviate neuronal damage, possibly as a result of growth factors’ modulation [43,44,45]. Moreover, activating GPR39 shows promise as a potential treatment approach for epilepsy [46]. However, the mechanisms underlying GPR39-mediated OAB treatment remain unclear.

To date, research has been limited regarding the effects of GPR 39 receptors and their agonist on bladder function in individuals with overactive bladder who also have mental health conditions.

Given that TC-G 1008 is the most commonly utilized ligand in GPR39 studies, our research aimed to assess its impact on both urinary bladder function and depressive symptoms. We employed an animal model of OAB induced by CORT for this evaluation. The complexity of our experiment stemmed from the critical roles that the urothelium and detrusor muscle play in regulating bladder function. Furthermore, OAB involves multiple neurotransmitter-related alterations that influence the control of the micturition reflex.

The CORT model of OAB and depression utilized in our current studies has proven to be a dependable framework for evaluating the potential of a given substance in treating OAB with concurrent depressive symptoms, as evidenced by the results of our earlier investigations. The foundation of this model rests on a widely accepted understanding that recurrent stress and overactivation of the hypothalamic–pituitary–adrenal (HPA) axis play significant roles in the onset and expression of depressive conditions. Studies conducted in laboratory settings have shown increased levels of cortisol in the plasma, urine, and cerebrospinal fluid of individuals with depression [47]. Research has also demonstrated that extended stress exposure or corticosterone administration affects the same cerebral regions (namely, the hippocampus, amygdala, and prefrontal cortex) that are impacted in patients suffering from depression [48]. Following the administration of CORT, we noted substantial alterations in the levels of the examined biomarkers across various tissues. These changes were evident in urine samples, the detrusor muscle, and urothelium of the bladder, as well as specific brain regions, namely, the hippocampus and prefrontal cortex. In line with our expectations, the animals subjected to 14 days of CORT exposure exhibited depressive-like behavior during the forced swim test (FST, developed by Porsolt et al. [49]). Additionally, these animals demonstrated changes in their overall locomotor activity (mobility) and displayed indicators of OAB, which were readily observed through a cystometric analysis. Significant increases were observed in their FNVC, DOI, AUC, and BP measurements, while VTNVC, VV, TP, and ICI levels showed a marked decrease. The findings of our investigation remained unaffected by alterations in rat locomotor activity, as no notable differences in overall movement were detected between the examined groups in our study or in the research conducted by Gregus et al. [49]. In our previous study [50], we validated the aforementioned FST using various medications. These included solifenacin and mirabegron, which primarily affect OAB through peripheral pathways, as well as duloxetine, which predominantly acts via the central pathway.

Furthermore, these debates about whether OAB causes affective disorders or vice versa indicate that depression and OAB might simply have overlapping pathological mechanisms [51]. In this scenario, corticotropin-releasing factor (CRF) might be a key player, as inhibiting its receptor (CRF1) could potentially alleviate symptoms associated with depression and overactive bladder (OAB) [4]. The changes in cytokines and neurotrophins observed in the hippocampus of rats during our experiment accurately mirrored those seen in depression. Chronic low-level inflammation is thought to be a contributing factor in the pathophysiology of depression. Consistent with our findings, the levels of pro-inflammatory cytokines (IL-1β, TNF-α, and IL-6) were elevated, while the levels of anti-inflammatory cytokines (IL-10 and TGF-β) were reduced in the hippocampus of the previously described rat model of depression [52]. Our study’s findings of reduced NGF and BDNF expression align with both clinical and animal research, which propose that depression is linked to neuronal degeneration in the hippocampus due to low neurotrophin levels [53].

As illustrated in Figure 1, our research revealed that the intraperitoneal injection of TC-G 1008 to activate GPR39 led to notable alterations in several critical variables. The compound TC-G 1008 enhanced urinary bladder function by positively affecting various cystometric measurements. It increased intercontraction interval (ICI), threshold pressure (TP), bladder capacity (BC), voided volume (VV), and volume threshold for non-voiding contractions (VTNVC). Additionally, it decreased the amplitude of non-voiding contractions (ANVC), frequency of non-voiding contractions (FNVC), area under the curve (AUC), basal pressure (BP), and detrusor overactivity index (DOI). These alterations in cystometric parameters indicate that TC-G 1008 successfully reduces the baseline tone and excitability of the detrusor muscle during the storage phase of micturition. Lastly, the FST, a behavioral assessment tool for rodents, was used to evaluate another aspect of TC-G1008’s antidepressant-like effects. This test allows researchers to measure the effectiveness of such treatments in animal models.

Well-known preclinical and clinical studies have shown antidepressant properties of substances that are antagonists of the glutamatergic system [11,54]. The GPR39 receptor, a TC- G 1008 agonist, has been recognized as a zinc-sensing receptor. Zinc, through its binding site on the NMDA receptor subunit, inhibits its activity in the central nervous system (CNS). The consequence of the activation of pathways associated with the GPR39 receptor is an increase in the transcription factor CREB, whose important role has been demonstrated in depression. CREB protein increases the level of brain-derived neurotrophic factor BDNF, which has a neuroprotective effect and is responsible for broadly understood neuroplasticity, which is extremely important in the therapeutic response to antidepressants [20].

The neurotrophic theory of depression suggests that reduced levels of brain-derived neurotrophic factor (BDNF) in individuals suffering from depression are crucial in the development of this mental health condition. Research has shown reduced BDNF levels in the brains of individuals who died by suicide [55], people with depression [56], and animals under stress [57], mirroring our findings. These diminished levels were particularly evident in the hippocampus and prefrontal cortex regions of the brain. In animal models of OAB and individuals diagnosed with OAB syndrome, the increased expression of NGF and BDNF is a well-documented occurrence, observed in both the bladder and the neural pathways [58].

Based on our current understanding, there is a lack of published evidence explaining how the GPR39 receptor directly influences NGF and BDNF expression in the bladder and urine following CORT stimulation.

Interestingly enough, TC-G 1008 treatment almost normalized the neurotrophin BDNF and NGF levels both in urine and examined brain areas. The administered treatment increased the previously lowered BDNF concentrations in the hippocampus and prefrontal cortex while reducing the elevated BDNF levels found in urine. Additionally, our experiment demonstrated that the GPR39 agonist counteracted the CORT-induced alterations in NGF and BDNF levels within the hippocampus and prefrontal cortex. However, it did not have a significant impact on the concentration of NGF in plasma.

Our previous research has shown that imipramine, a common antidepressant, may be effective in treating dry overactive bladder (OAB). In rats exposed to 13-cis-retinoic acid, this drug reversed several cystometric parameters associated with detrusor overactivity (DO). Additionally, it significantly reduced elevated levels of CRF in the plasma, hippocampus, and amygdala [59]. Doxepin, an alternative tricyclic antidepressant, was found to decrease both the frequency of nighttime urination and instances of nocturnal incontinence when administered at 50–75 mg [60]. Furthermore, our prior research findings indicated that blocking CRF1 receptors with SN003 [59,61], suppressing Rho kinase activity using GSK 269962 [62], and inhibiting myosin II through blebbistatin [63] enhanced both urodynamic measurements and mood-related behaviors in an experimental model combining detrusor overactivity and depression.

CRF is found in both peripheral tissues and the central nervous system, with significant concentrations in brain areas like the hypothalamus and amygdala [64,65]. This substance is notably present in regions responsible for controlling urination, specifically in the neurons of Barrington’s nucleus that extend to bladder motor neurons and in the locus coeruleus [66]. These connections suggest a bladder–brain relationship, with CRF acting as a key component in this network. Elevated CRF concentrations were observed in the cerebrospinal fluid of individuals who died by suicide [67] and those experiencing depression [68,69,70].

The elevated CRF levels noted in the CORT-treated group during our study may be attributed to the HPA axis disruption resulting from the 14-day CORT administration protocol. Research has established that acute stress triggers the release of CRF, which subsequently stimulates the production of adrenocorticotropic hormone (ACTH). This hormone, in turn, enhances the synthesis of glucocorticoids within the adrenal cortex. The release of cortisol in humans and CORT in rodents operates through a well-documented negative feedback mechanism, inhibiting the further secretion of both CRF and ACTH. Extended periods of stress disrupt the normal operation of the HPA axis, resulting in a glucocorticoid-mediated rise in CRF production. Many psychiatric-related conditions, including depression, have been linked to HPA axis dysfunction, characterized by increased secretion of CRF and ACTH [71].

Additional studies on biomarkers in the bladder detrusor muscle showed that TC-G 1008 reduced the elevated levels of both VAChT and Rho kinase caused by CORT. The RhoA/Rho-kinase signaling pathway is thought to be linked to depression, and glucocorticosteroids have been found to affect its expression [72]. Bladder basal tone is also influenced by the RhoA/Rho kinase signaling pathway, and activating this pathway may result in increased unintentional bladder contractions [73]. Furthermore, the elevated expression of VAChT, potentially triggered by BDNF, suggests an enhancement in cholinergic neurotransmission [62,74].

Significantly, our study’s findings revealed that TC-G 1008, administered at 15 mg/kg/day, did not negatively impact MVP or PVR (Figure 1). This twofold effect showcased imperatorin’s capacity to enhance the storage phase while sustaining voiding efficiency. Furthermore, TC-G 1008’s direct influence on detrusor stability wsa evidenced by its capacity to decrease DOI, a widely used indicator in in vivo research for diagnosing overactive bladder, as unstable detrusor contractions typically manifest during the storage phase in OAB [75].

Additionally, TC-G 1008 showed the capability to enhance VTNVC, as illustrated in Figure 1. In animal studies using cystometry, VTNVC functions as a comparable measurement to the volume at which the initial involuntary detrusor contraction occurs in human subjects. VTNVC is considered a highly dependable indicator for evaluating the effectiveness of OAB treatments, as it correlates with a decrease in the number of urinary incontinence episodes and a reduction in micturition frequency [76].

For the subsequent phase of our study, we examined the biomarkers present in the bladder urothelium following the simultaneous administration of CORT and TC-G 1008. Our findings revealed that CORT led to increased levels of various substances, including CGRP, ATP, CRF, OCT-3, and TRPV1. Additionally, we investigated pro-inflammatory cytokines, specifically TNF-α, IL-1β, IL-6, and IL-10. Additionally, we noted TC-G 1008’s ability to mitigate these CORT-induced effects in all cases examined. In unmyelinated C nerve fibers located within the urothelium and suburothelium, CGRP and TRPV-1 were found to be co-expressed [77]. These components are believed to contribute to the bladder’s phasic activity through localized axonal reflexes [78]. Patients with OAB exhibit an elevated concentration of CGRP immunoreactive fibers (Figure 4), indicating their potential involvement in the condition’s underlying mechanisms [79]. Additionally, individuals with sensory OAB show heightened levels of TRPV1 mRNA expression [80]. The elevated levels of CGRP and TRPV-1 indicate the activation of afferent C fibers [81], potentially contributing to the cystometric abnormalities observed in our research. The non-neuronal release of ACh from the urothelium is facilitated by OCT3 [82]. The efficacy of TC-G 1008 in reducing acetylcholine (ACh) release, a key component of overactive bladder (OAB) treatment, is evidenced by reduced expression of both vesicular acetylcholine transporter (VAChT) and organic cation transporter 3 (OCT3). This study corroborates our earlier findings, which demonstrated that female Wistar-Kyoto rats with spontaneous hypertension and the retinyl acetate-induced model of detrusor overactivity (DO)—a condition diagnosed in 60–90% of OAB patients—exhibited comparable alterations in calcitonin gene-related peptide (CGRP), OCT3, and VAChT levels, as documented in the research by Wróbel et al. [33].

Bladder regulation is a multifaceted process involving various stages. Our research sought to investigate how imperatorin affected c-Fos expression in key neuronal centers associated with urination, specifically the MPA, PMC, and vlPAG (illustrated in Figure 7). The expression of c-Fos, which indicates neuronal activity [83], is crucial for comprehending how the PMC and PAG contribute to the supraspinal regulation of continence and urination [84]. In cases of OAB, stimulation of the bladder activates key micturition areas in the brain, leading to an increase in c-Fos expression, as noted by Kim et al. [85]. Furthermore, their study revealed a notable increase in c-Fos expression within the neuronal voiding centers of the OAB animal model [85]. Our present investigation demonstrated that RA triggered substantial c-Fos expression across all examined centers; nevertheless, this effect was reduced by IMP (Figure 7).

Subsequently, our attention shifted to examining the expression of ATP, which plays a crucial role in numerous phases of molecular metabolic processes. This expression increased after CORT installation and subsequently decreased after TC-G 1008 administration. Sensory nerves, interstitial cells, and smooth muscle are influenced by ATP. Bladder dysfunction is closely associated with irregularities in ATP production or release [86]. ATP is considered a potential biomarker for OAB due to its ability to induce detrusor overactivity. Furthermore, recent studies indicate that abnormal regulation of extracellular ATP plays a role in the pathophysiology of depression [87].

The substantial decline in a patient’s quality of life due to OAB symptoms can create stress, potentially triggering the onset of mood disorders [30]. Conversely, stress-related depression or the use of antidepressants may contribute to the development of new-onset OAB [31,88]. The common occurrence of OAB and depression may also be attributed to the involvement of inflammation in the underlying mechanisms of both conditions.

Research has indicated that oxidative stress may play a role in the development of lower urinary tract symptoms. While copper has been shown to promote oxidative stress, zinc is known to be involved in protecting against it [89]. Research by Prasad et al. [90] utilizing cell cultures demonstrated that elevated zinc levels in cells led to enhanced A20 protein expression and diminished activation of IκB kinase (IKK)-α/NF-κB signaling and pro-inflammatory cytokines. Additionally, Bao et al. [91] observed that zinc supplementation decreased the concentrations of inflammatory molecules such as interleukin 6 (IL-6), TNF-α, monocyte chemoattractant protein (MCP-1), C-reactive protein, intercellular adhesion molecule 1 (ICAM-1), and E-selectin. Furthermore, it increased the expression of peroxisome proliferator-α (PPAR-α) and the anti-inflammatory protein A20.

Most research, including our own previous investigations, suggests that individuals with overactive bladder (OAB) exhibit increased concentrations of inflammatory biomarkers in their serum or urine, such as TNF-α [92]. In a similar vein, patients diagnosed with various forms of depression often show significantly elevated levels of IL-1β in their cerebrospinal fluid (CSF), serum, or urine [93], as well as higher TNF-α concentrations in their serum [94], when compared to non-depressed individuals. Our experimental findings revealed that rats treated with CORT exhibited indicators of inflammation and alterations in neurotrophic factors. These observations are characteristic of both OAB and depression, corroborating our previous results. IL-10’s self-regulating function and its capacity to suppress the production of other cytokines may underscore its importance as an inhibitor of inflammatory processes. Our research revealed significantly reduced IL-10 levels in patients with overactive bladder (OAB) compared to healthy controls, suggesting that these individuals may not have generated an adequate IL-10 response to counteract the inflammation associated with OAB. This insufficient production of IL-10 could potentially allow inflammation to persist and expand, thereby contributing to the development of OAB.

Recent human studies have provided growing evidence linking the increased production of cytokines, both peripherally and centrally, to the onset of mood disorders [95,96]. The heightened inflammatory response subsequently contributes to the abnormal activation of the glutamatergic system, resulting in excitotoxicity and reduced neurotrophic support within the central nervous system. Indeed, the pathology of depression is associated with disruptions in the immune, monoaminergic, and glutamatergic systems.

Inflammation has important physiological effects on mood and behavior. The hypothesis is that altered glutamate neurotransmission links inflammation and depression, which may be due in part to the alteration of TRP metabolism via KP and its metabolites [97]. Kynurenines exhibit a diverse array of contrasting biological effects, ranging from cytotoxic to cytoprotective, oxidative to antioxidant, and pro-inflammatory to anti-inflammatory. Furthermore, the specific role of a given metabolite can be influenced by an intricate system of internal factors and external modifiers, such as the availability of TRP, inflammatory conditions, and exposure to environmental toxicants [24].

The immune system plays a role in regulating KP; pro-inflammatory cytokines trigger the activation of step-limiting enzyme, IDO [98]. IDO then converts TRP to KYN, with the [KYN]/[TRP] ratio serving as an indicator of this enzymatic activity. KYN is subsequently transported across the blood–brain barrier through an active process [99]. In both peripheral and central regions, additional enzymes further break down KYN, producing various metabolites including QA and KYNA. These resulting compounds are neurologically active, influencing glutamate signaling at NMDA glutamate receptors. QA displays a high affinity for the NMDA receptor’s glutamate binding site, while KYNA blocks the NMDA receptor glycine site. An altered proportion between produced in situ QA and KYNA may result in a pathological stimulation of the NMDA receptor [100]. In addition, chronically increased pro-inflammatory cytokine levels may lead to TRP depletion and reduced serotonin levels, which can strengthen depressive symptoms [101].

In mice, an induction of peripheral inflammation elevates the levels of KYN and QA in the brain, with subsequent depression-like symptoms [100]. Inhibitors of IDO may alleviate these changes [102].

Research on human subjects has revealed variations in KP metabolite levels between individuals with depression and healthy controls [98,103,104]. This suggests that the KP might play a significant role in the development of depression. Individuals suffering from depression exhibit a reduced serum [KYNA]/[QA] ratio [104], indicating the possibility of enhanced NMDA receptor activation in these patients. Additionally, research has revealed that people who have attempted suicide have higher concentrations of QA in their cerebrospinal fluid compared to control subjects. In individuals who have attempted suicide, elevated QA levels in cerebrospinal fluid were found to be linked with higher overall scores on the suicide intent scale. Additionally, decreased KYNA concentrations were found to be associated with more intense symptoms of depression [98]. Elevated QA was reported in the cingulate cortex of suicide victims with depression [105].

Here we evaluated the link between CORT-induced OAB and depression and KYNA measured in plasma and selected brain areas. As expected, we observed a significant decrease in KYNA concentrations after the administration of CORT in the plasma, hippocampus, and prefrontal cortex. Furthermore, we revealed that intraperitoneal administration of TC-G 1008 reversed changes in KYNA levels. It seems of interest to perform clinical studies aimed at analyzing peripheral KYNA levels in relation to OAB symptoms and potential treatment with a selective GPR39 receptor agonist.

To summarize, this research examined the potential of TC-G 1008 as a therapeutic approach for OAB in individuals experiencing depressive symptoms. The findings provided promising results supporting its effectiveness as a treatment option. The results of our study demonstrated that the GPR39 agonist, when administered intraperitoneally at 15 mg/kg/day, significantly reduced the pathological alterations in bladder function associated with detrusor overactivity. TC-G 1008 enhanced various cystometric measurements, including voided volume, threshold pressure, and bladder capacity. Additionally, it decreased both the frequency and magnitude of non-voiding contractions and bladder pressure. The observed outcomes indicate that TC-G 1008 enhances the tone and excitability of the detrusor muscle during the storage phase of micturition. Biochemical studies showed that TC-G 1008 decreased the concentrations of mediators linked to OAB and psychiatric conditions, indicating that this compound might influence inflammatory processes and neural communication pathways related to OAB and depression. Furthermore, TC-G 1008 was found to diminish c-Fos expression in brain regions controlling urination, suggesting a possible mechanism for its effectiveness in alleviating OAB symptoms. The interplay between neurotransmitter systems, particularly the GABAergic and glutamatergic pathways, plays a critical role in regulating the micturition reflex, and may underlie some of the observed effects of TC-G 1008. Recent evidence has highlighted the importance of GABAergic and glutamatergic plasticity in key brain regions, such as the medial prefrontal cortex, which are central to both bladder control and mood regulation [106]. Additionally, the modulation of excitatory and inhibitory GABA signals is essential for maintaining the balance between excitatory and inhibitory impulses that govern the micturition reflex [107,108]. The involvement of GPR39 in these pathways through mechanisms such as zinc-mediated neurotransmission and neuroplasticity provides further support for its therapeutic potential in addressing the overlapping pathophysiology of OAB and depression. These insights reinforce the need for further research into how to target GPR39 and its downstream pathways, which could advance treatment strategies for OAB with comorbid depression.

In the current literature, there is no developed method of treating patients with depression and OAB. Moreover, there are no recommendations for this type of treatment of patients. Therefore, our studies indicate the need for further clinical research on the modulation of GPR39 receptor in the combined therapy of depression and OAB. Furthermore, the GPR-39 agonist’s favorable safety profile, demonstrated by the lack of negative impacts on urinary function and its neuroprotective qualities, gives hope that it may become a potential treatment for OAB patients.

Nevertheless, there are some limitations of our study. The experiments were performed in female rats, thus limiting the applicability of the results across both sexes. Although this study offers valuable knowledge about TC-G 1008’s mechanism in the bladder, additional exploration is necessary to clarify the involvement of specific intracellular pathways. Additional research is needed to evaluate the long-term safety and effectiveness of TC-G 1008 in human clinical studies.

## 4. Materials and Methods

The Local Ethics Committee authorized all implemented protocols (number 323TR/22), which were conducted in compliance with the applicable European legislation governing experimental research on animal subjects.

### 4.1. Animals

The experiments utilized 48 female Wistar rats with initial weights ranging from 200 to 225 g. Each rat was housed separately in metabolic cages (3700M071, Tecniplast, West Chester, PA, USA) within rooms that had controlled environmental conditions. These conditions included a temperature of 22–23 °C, a natural light/dark cycle, and relative humidity of approximately 45–55%. The rats were provided unrestricted access to food and water throughout the study. The experiment involved 48 rats divided equally into four groups, each containing 12 animals. These groups were: 1. CON: received a placebo for 21 days (14 days + 7 days); 2. CORT: administered corticosterone (20 mg/kg/day) for 14 days, followed by a placebo for 7 days; 3. GPR: given a placebo for 14 days, then the GPR39 agonist TC-G 1008 (15 mg/kg/day) for 7 days; 4. CORT + GPR: treated with corticosterone (20 mg/kg/day) for 14 days, then the GPR39 agonist for 7 days. The rats were assigned to these groups randomly and had no prior experimental exposure. Each animal underwent testing only once.

### 4.2. Drugs

The experiment utilized the following compounds: CORT ((11β)-11,21-Dihydroxypregn-4-ene-3,20-dione), obtained from Tocris Bioscience in Bristol, Great Britain, was administered subcutaneously at 20 mg/kg daily for a two-week period. Additionally, TC-G 1008 (N-[3-Chloro-4-[[[2-(methylamino)-6-(2-pyridinyl)-4-pyrimidinyl]amino]methyl]phenyl]methanesulfonamide), also sourced from Tocris Bioscience in Bristol, Great Britain, was employed as a highly effective and specific GPR39 agonist, with EC50 values of 0.4 and 0.8 nmol/L for rat and human receptors, respectively. The compound TC-G 1008 was injected into the peritoneal cavity at a daily dosage of 15 mg/kg over a 7-day period. The dosages of the administered substances were chosen based on findings from our earlier studies and published data and were verified and adjusted in preliminary tests conducted in our laboratory [109,110]. Animals in the control group received an equivalent volume of vehicle injection.

### 4.3. Surgical Procedures

The surgical interventions were conducted using previously established methods [61], with anesthesia administered through intraperitoneal injection. The anesthetic mixture consisted of ketamine hydrochloride (Ketanest; Pfizer, Inc., New York, NY, USA) at a dose of 75 mg/kg and xylazine (Sedazin; Biowet, Puławy, Poland) at 15 mg/kg. The rats were positioned on their backs on a heated pad maintained at 37 °C. Anesthesia was deemed sufficient when the animals exhibited no spontaneous movement and failed to respond to a painful pinch of their toes. A vertical midline incision of approximately 10 mm was made to access the shaved and cleaned abdominal area. The bladder was carefully separated from surrounding tissues. A polyethylene catheter with two lumens (internal diameter: 0.28 mm, external diameter: 0.61 mm; BD, Franklin Lakes, NJ, USA) was prepared by filling it with physiological saline and attaching a cuff at one end. This catheter was then inserted into the bladder’s dome through a small incision and secured using a 6-0 Vicryl suture. The catheter was implanted beneath the skin and emerged in the area behind the shoulder blades, where it was joined to a plastic connector to prevent the animal from dislodging it. During the same procedure, a polyethylene catheter (inner diameter 0.28 mm, outer diameter 0.61 mm; BD) was inserted into the carotid artery to monitor blood pressure. This catheter was filled with heparinized physiological saline at a concentration of 40 IU/mL. The catheters were implanted beneath the skin and brought out in the area behind the shoulder blades. They were then attached to a plastic connector to prevent the animal from dislodging them. To minimize the formation of adhesions, 0.85 mL of Healon (Pharmacia A.B., Uppsala, Sweden) was applied around the urinary bladder as a final step. Multiple layers were used to close the abdomen. The anatomical layers were sutured with 4/0 catgut. Silk ligatures were employed to seal the free ends of the catheters. To prevent urinary tract infection, the animals received a subcutaneous injection of 100 mg cefazolin sodium hydrate (Biofazolin; Sandoz, Warsaw, Poland).

### 4.4. Conscious Cystometry

Three days following the final TCS injection, cystometric studies were conducted on awake, freely moving rats, utilizing protocols previously described in the literature [62,111]. A three-way stopcock was used to link the bladder catheter to two devices: a pressure transducer (FT03; Grass Instruments, West Warwick, RI, USA) positioned at bladder level, and a microinjection pump (CMA 100, Microject, Solna, Sweden). This setup allowed for the measurement of intravesical pressure and the infusion of physiological saline into the bladder. A controlled cystometry procedure was conducted by gradually introducing physiological saline into the bladder at a steady rate of 0.05 mL/min (equivalent to 3 mL/h) and at room temperature (22 °C). This process was carried out to induce repeated voiding while the subject remained awake. Pilot studies determined that an infusion rate ranging from 0.05 to 0.1 mL/min produced bladder cystometry profiles in rats that were comparable to those observed in their intact lower urinary tract. This rate was subsequently used for the infusion. Typically, increasing the rate of infusion resulted in either an expansion of bladder volume or a spontaneous contraction of the bladder’s detrusor muscle. The signal from the pressure transducer, in analog form, was enhanced and converted to digital format using the Polyview system, manufactured by Grass Instruments in West Warwick, RI, USA. The measurement of urine output was conducted using a fluid collection device connected to a force displacement transducer (FT03C; Grass Instruments, West Warwick, RI, USA). Both transducers were linked to a polygraph (7 DAG, Grass Instruments, West Warwick, RI, USA) for data recording. A Grass polygraph (Model 7E, Grass Instruments, West Warwick, RI, USA) was utilized to continuously record cystometry profiles and micturition volumes, which were then graphically determined. Data analysis was conducted using a sampling frequency of 10 samples/s. The reported measurements for each animal represent the mean of five bladder voiding cycles after achieving consistent urination patterns. To ensure unbiased results, all procedures were carried out by an individual who was unaware of the treatment conditions.

### 4.5. Biochemical Analyses

Various biomarkers were evaluated in the hippocampus, prefrontal cortex, and plasma of rats. These included interleukin-1β (IL-1β; Cloud-Clone, Katy, TX, USA, SEA563Ra), interleukin-6 (IL-6; LifeSpanBioSciences, Seattle, WA, USA, LS-F25921-1), interleukin-10 (IL-10, Cloud-Clone, XX, LS-F25921-1), tumor necrosis factor α (TNF-α, LifeSpanBioSciences, Seattle, WA, USA, LS-F5193), corticotropin-releasing factor (CRF, Alpco, Salem, NH, USA, CN 48-CRFMS-E01), brain-derived neurotrophic factor (BDNF, PROMEGA, Walldorf, Germany, CN G7610), nerve growth factor (NGF, LifeSpanBioSciences, Seattle, WA, USA, CN LS-F25946-1), and KYNA (ELK Biotechnology, Denver, CO, USA, ELK9739).

The bladder urothelium was analyzed to determine the concentrations of several biomarkers. These included Calcitonin Gene Related Peptide (CGRP; Biomatik, Kitchener, ON, Canada, CN EKU02858), organic cation transporter 3 (OCT3, antibodies-online, Limerick, PA, USA, CN ABIN6227163), transient receptor potential cation channel subfamily V, member 1 (TRPV1, LSBio, Poznań, Poland, LS-F36019), and ATP citrate lyase (ATP, LifeSpanBioSciences, Seattle, WA, USA, LS-F10730).

In the bladder detrusor muscle, we identified the presence of vesicular acetylcholine transporter (VAChT, LifeSpanBioSciences, Seattle, WA, USA, CN LS-F12924-1) and Rho kinase (ROCK1, LifeSpanBioSciences, Seattle, WA, USA, LS-F32208). Additionally, we measured the levels of nerve growth factor (NGF, LifeSpanBioSciences, Seattle, WA, USA, CN LS-F25946-1) and brain-derived neurotrophic factor (BDNF, PROMEGA, Walldorf, Germany, CN G7610) in urine samples.

The expression of c-Fos (c-Fos, MyBioSource, San Diego, CA, USA, MBS729725) was evaluated in key micturition regions, including the medial preoptic area (MPA), ventrolateral periaqueductal gray (vlPAG), and pontine micturition center (PMC).

All analyses were conducted in accordance with the manufacturer’s protocols, with each sample tested in duplicate. The findings are reported in pg/mL units.

### 4.6. CRF Measurement

To avoid the diurnal fluctuation in the HPA axis hormone levels, CORT/salt was administered between 8 a.m. and 9 a.m. Following the conclusion of the behavioral experiments, we extracted blood samples and dissected the hypothalamus and prefrontal cortex. These brain regions were identified using a rat brain stereotaxic atlas, with the large crown serving as a reference point [112]. These brain structures were homogenized [113]. The concentration of CRF in the hypothalamus was assessed according to the manufacturer’s instructions using a high-sensitivity commercial enzyme immunoassay (LBS) (CRF, Alpco, Salem, NH, USA, CN 48-CRFMS-E01).

The brains were swiftly extracted and placed in chilled saline (2–8 °C). The hippocampus was identified and removed on a cooled surface using the rat brain stereotactic atlas. It was then promptly frozen with dry ice and kept at –80 °C until analysis. The sample was mixed with an extraction buffer (10 mmol/L PBS (pH 7.2) with 0.2% Nonidet P-40) and homogenized in an ice bath. After centrifugation (18,360× *g*, 20 min) at 4 °C, the resulting supernatants were extracted and immediately subjected to a biochemical analysis.

### 4.7. Determining the Expression Levels of c-Fos in Central Micturition Areas

The hypothalamus and prefrontal cortex were extracted using the rat’s brain stereotactic atlas, with the bregma serving as the reference point [114]. From each rat, an average of ten sections per region were collected.

### 4.8. Forced Swim Test

The forced swim test was conducted following the methodology described by Porsolt et al. [115]. Individual rats were introduced into cylindrical glass containers (65 cm in height, 25 cm in diameter) filled with 48 cm of water at a temperature between 23 and 25 °C. Following a 15-min immersion in water (pre-test), the subjects were transferred back to their original enclosures. The next day, 24 h after the initial forced swim, the rats underwent a 5-min retest under identical swimming conditions. The retest sessions were recorded on video from the side of the cylinders. A person unaware of the treatment condition evaluated the footage using a behavioral sampling method. Immobility in the rat was determined when it floated passively, making only minimal movements to keep its head above water.

### 4.9. Locomotor Activity

Animal movement patterns were evaluated using a Digiscan device, specifically an Optical Animal Activity Monitoring System manufactured by Omnitech Electronics, Inc. in Columbus, OH, USA. In a room illuminated by soft red lighting, transparent acrylic open field boxes served as activity chambers. Animal movement was tracked using the Digiscan system, which employed a network of invisible infrared beams. These beams were arranged in a uniform grid pattern across the animal enclosure. The animal’s position was detected when its body disrupted the beams in the Digiscan. Each beam interruption was logged as an activity score. The OMNIPRO data collection program received and stored cumulative counts every 15 min. Before conducting behavioral analysis, participants were allowed to acclimate in activity chambers for 15 min. The experiments took place in a room that was soundproof. The study measured horizontal movement. The measurement involved counting the total number of horizontal sensor beam interruptions over a 1 h period. To ensure unbiased results, all procedures were conducted by an individual who was unaware of the treatment assignments.

### 4.10. Study Design

Three days following the final GPR39 agonist injection, researchers conducted a series of studies: cystometry, the Porsolt test, and locomotor activity assessment. Upon completion of the cystometric and behavioral evaluations, the subjects were euthanized via decapitation, and their brain and urinary bladder tissues were harvested for further analysis.

### 4.11. Statistical Analysis

Statistical analysis was conducted using one-way ANOVA, followed by Tukey’s post hoc test. Results were expressed as means ± SD. Statistical significance between groups was determined at *p* < 0.05.

## Figures and Tables

**Figure 1 ijms-25-12630-f001:**
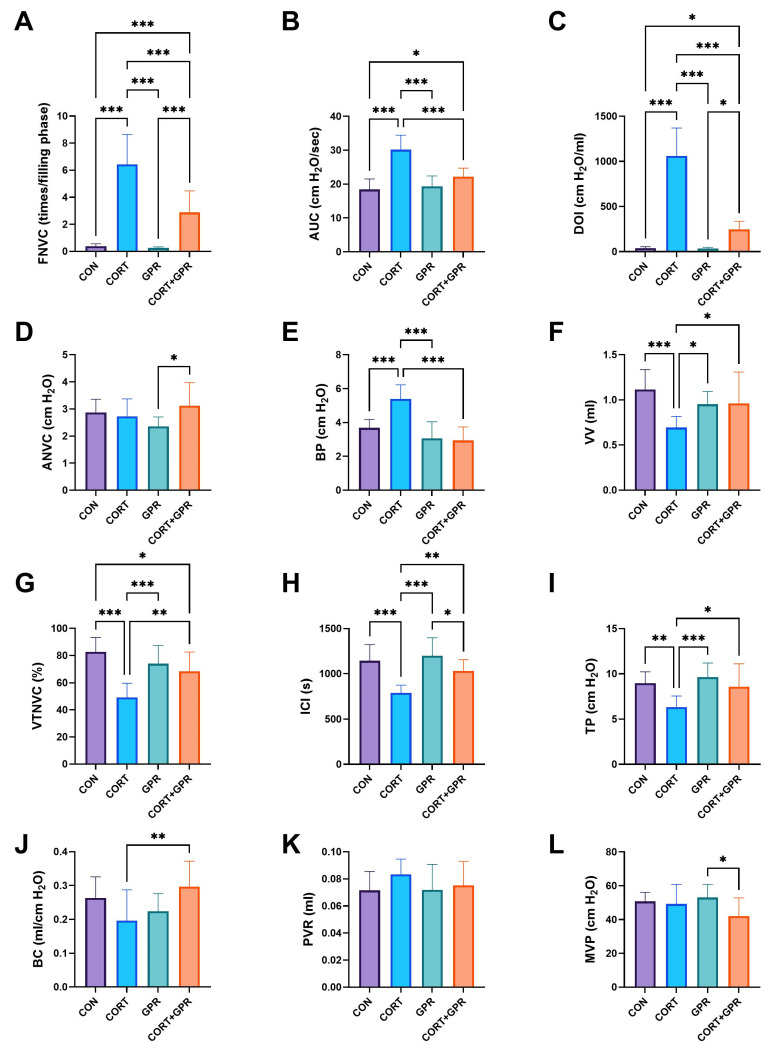
Effect of TC-G 1008 on conscious cystometry parameters in the corticosterone-induced overactive-bladder rat model. (**A**) Non-voiding contraction amplitude (FNVC); (**B**) area under the pressure curve (AUC); (**C**) detrusor overactivity index (DOI); (**D**) non-voiding contraction frequency (ANVC); (**E**) basal pressure (BP); (**F**) voided volume (VV); (**G**) volume threshold to elicit non-voiding contractions (VTNVC); (**H**) intercontraction interval (ICI); (**I**) threshold pressure (TP); (**J**) bladder compliance (BC); (**K**) post-void residual (PVR); (**L**) micturition voiding pressure (MVP). Data presented as means ± SD, (n = 12 mice per group), * *p* < 0.05, ** *p* < 0.01, *** *p* < 0.001.

**Figure 2 ijms-25-12630-f002:**
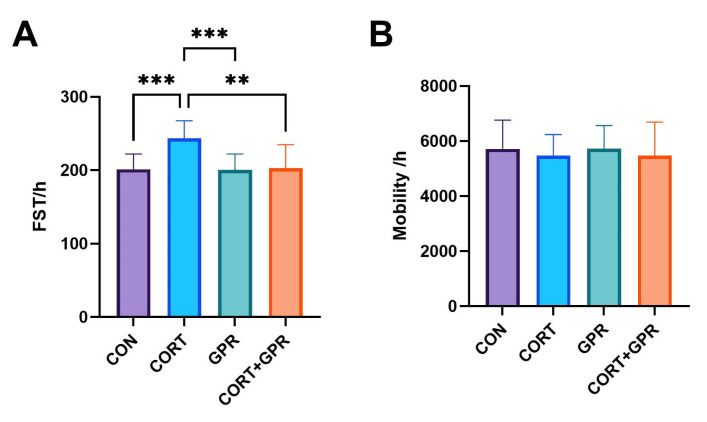
Effect of TC-G 1008 on forced swim test and locomotor activity in the corticosterone (CORT)-induced depression model in rats. (**A**) Immobility time in the forced swim test (FST) following 14 days of treatment with CORT (20 mg/kg/day) and the reversal effect by TC-G 1008 (15 mg/kg/day). (**B**) Locomotor activity (mobility) assessed after treatment with TC-G 1008 (15 mg/kg/day) alone or in combination with CORT. Data are presented as means ± SD, (n = 12 rats per group), ** *p* < 0.01, *** *p* < 0.001.

**Figure 3 ijms-25-12630-f003:**
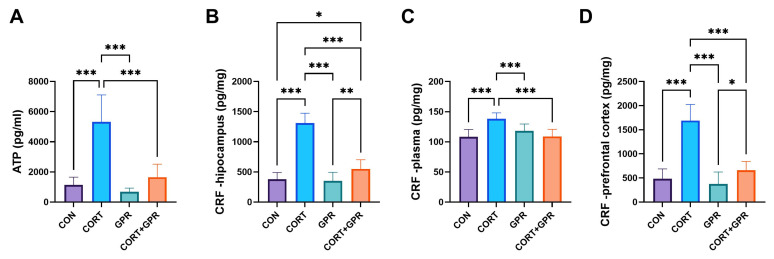
Effect of administration of TC-G 1008 (15 mg/kg/day) on kynurenic acid levels, respectively, in (**A**) adenosine triphosphate (ATP) and corticotropin-releasing factor (CRF), respectively, in (**B**) hipocampus, (**C**) plasma, (**D**) prefrontal cortex of rats subjected to CORT treatment (20 mg/kg/day, s.c.). Data are presented as means ± SD, (n = 12 rats per group), * *p* < 0.05, ** *p* < 0.01, *** *p* < 0.001.

**Figure 4 ijms-25-12630-f004:**
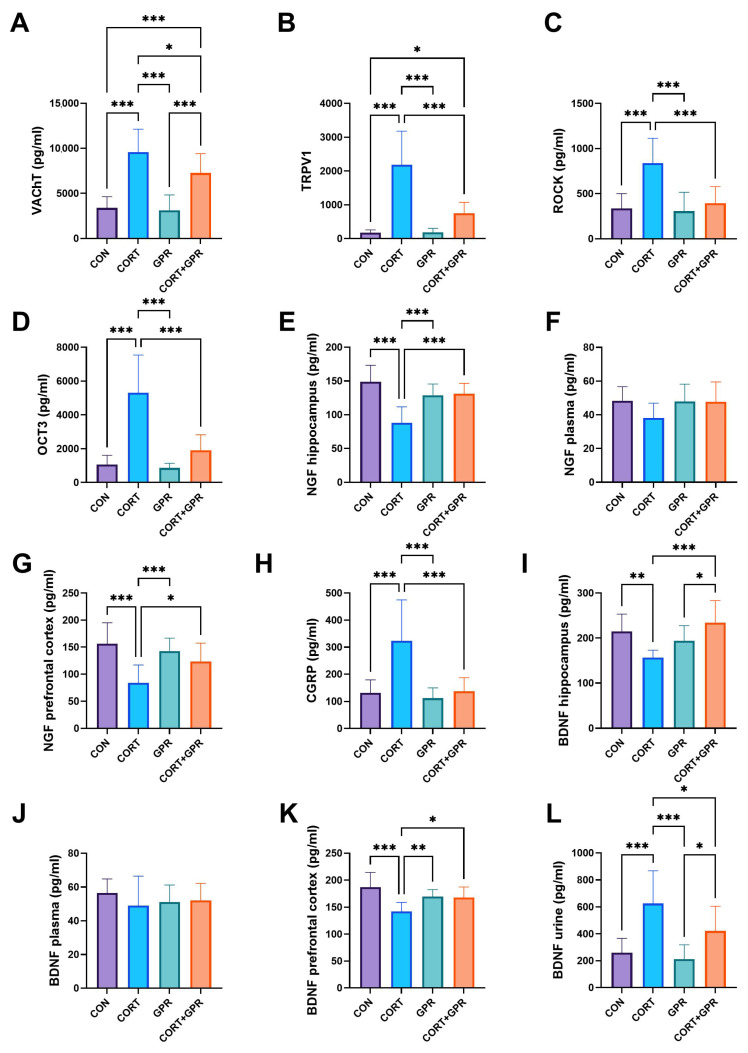
Effect of administration of TC-G 1008 (15 mg/kg/day) on selected biochemical parameters: (**A**) vesicular acetylcholine transporter (VAChT); (**B**) transient receptor potential cation channel subfamily V, member 1 (TRPV1); (**C**) Rho kinase (ROCK); (**D**) organic cation transporter 3 (OCT3); (**E**) nerve growth factor (NGF) in hippocampus; (**F**) NGF in plasma; (**G**) NGF in prefrontal cortex; (**H**) calcitonin gene-related peptide (CGRP); (**I**) brain-derived neurotrophic factor (BDNF) levels in the hippocampus; (**J**) BDNF plasma levels; (**K**) BDNF levels in the prefrontal cortex; (**L**) BDNF levels in the urine of rats subjected to CORT (20 mg/kg/day, s.c.). Data are presented as means ± SD, (n = 12 rats per group), * *p* < 0.05, ** *p* < 0.01, *** *p* < 0.001.

**Figure 5 ijms-25-12630-f005:**
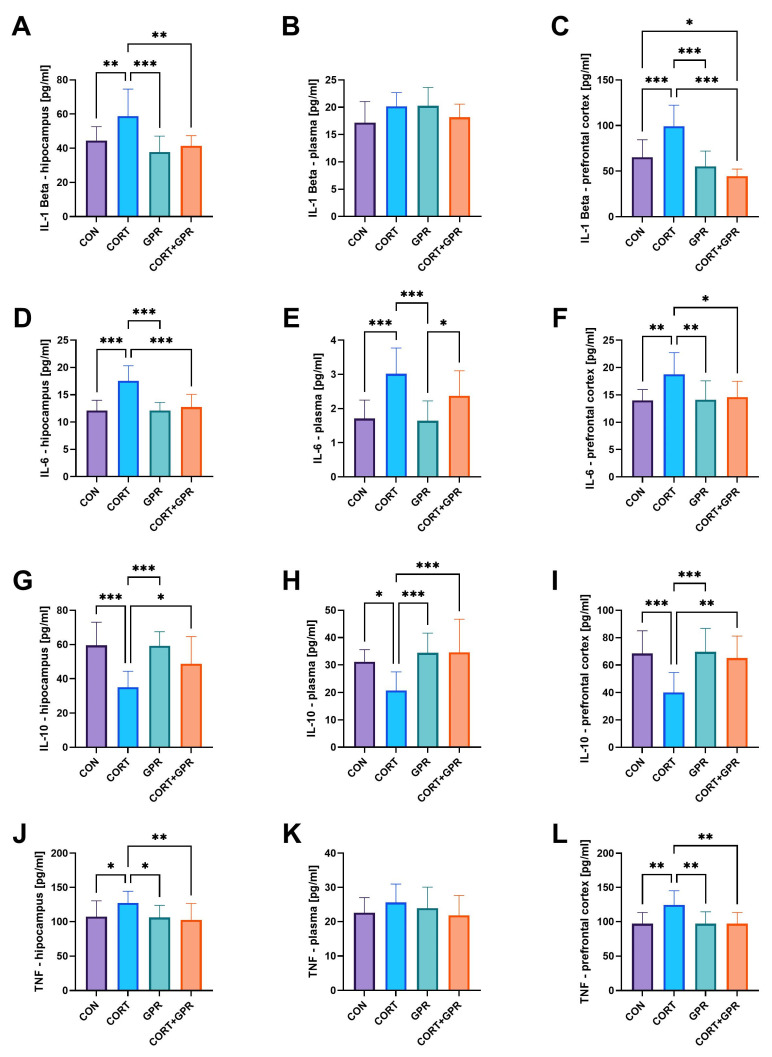
Effect of administration of TC-G 1008 (15 mg/kg/day) on interleukin 1-β (IL-1 Beta). Respectively. In the (**A**) hippocampus, (**B**) plasma, (**C**) prefrontal cortex; interleukin-6 (IL-6), respectively, in the (**D**) hippocampus, (**E**) plasma, (**F**) prefrontal cortex; interleukin-10 (IL-10), respectively, in the (**G**) hippocampus, (**H**) plasma, (**I**) prefrontal cortex; and tumor necrosis factor α (TNF-α) levels in the (**J**) hippocampus, (**K**) plasma, (**L**) prefrontal cortex of rats given CORT (20 mg/kg/day, s.c.). Data are presented as means ± SD, (n = 12 rats per group), * *p* < 0.05, ** *p* < 0.01, *** *p* < 0.001.

**Figure 6 ijms-25-12630-f006:**
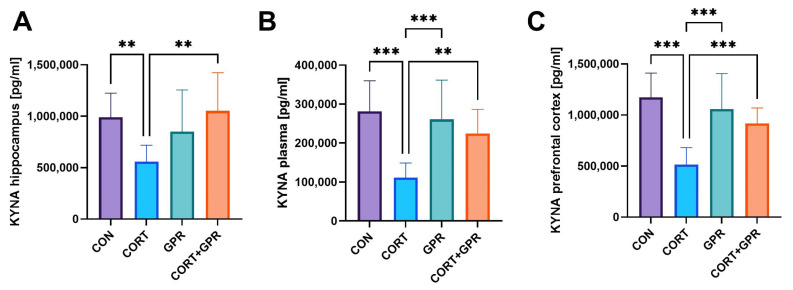
Effect of administration of TC-G 1008 (15 mg/kg/day) on kynurenic acid (KYNA) levels, respectively, in the (**A**) hippocampus, (**B**) plasma, (**C**) prefrontal cortex of rats subjected to CORT treatment (20 mg/kg/day, s.c.). Data are presented as means ± SD, (n = 12 rats per group), ** *p* < 0.01, *** *p* < 0.001.

**Figure 7 ijms-25-12630-f007:**
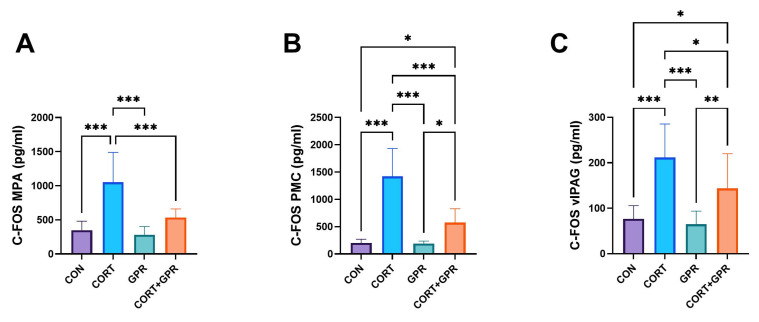
Effect of TC-G 1008 (15 mg/kg/day) on c-Fos expressions in the neuronal voiding centers: (**A**) medial preoptic nucleus (MPA), (**B**) pontine micturition center (PMC), and (**C**) ventrolateral periaqueductal gray (vlPAG) after the induction of overactive bladder with corticosterone (CORT) (20 mg/kg/day, s.c.). Data are presented as means ± SD, (n = 12 rats per group), * *p* < 0.05, ** *p* < 0.01, *** *p* < 0.001.

## Data Availability

The raw data supporting the conclusions of this article will be made available by the authors on request.

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
