# Peer review of "The GPR39 Receptor Plays an Important Role in the Pathogenesis of Overactive Bladder and Corticosterone-Induced Depression"

_ijms, 2024, doi:10.3390/ijms252312630_

Round 1

Reviewer 1 Report

Comments and Suggestions for Authors

The presentation of a GPR39 agonist as exerting effects related to OAB in female rats is attractive and clear. I suggest edition to improve relevance.

a) Please highlight the possible mechanisms involved in the observed effects, albeit if not there were included on the measured variables. In this sense, I hope edition in introduction and discussion regarding potential indirect effects in Autonomous Nervous System and the GABA-Glu balance.

b) Consider the content of the following articles to edit and generate a deeper discussion in regards of the potential mechanisms related with the effects in bladder.

Fogaça, M. V., Wu, M., Li, C., Li, X. Y., Duman, R. S., & Picciotto, M. R. (2023). M1 acetylcholine receptors in somatostatin interneurons contribute to GABAergic and glutamatergic plasticity in the mPFC and antidepressant-like responses. Neuropsychopharmacology48(9), 1277-1287.

Yoshimura, N., Miyazato, M., Kitta, T., & Yoshikawa, S. (2014). Central nervous targets for the treatment of bladder dysfunction. Neurourology and Urodynamics33(1), 59-66.

Kontani, H., & Ueda, Y. (2005). A method for producing overactive bladder in the rat and investigation of the effects of GABAergic receptor agonists and glutamatergic receptor antagonists on the cystometrogram. The Journal of urology173(5), 1805-1811.

c) Please add the meaning of the labels used in the charts. Albeit if it is predictable. Check the charts legends for the use of adequate asterisks meaning. For example, in Fig. 2, the *p=0.05 is in the legend, but no presented in the chart.

d) Check the use of a the neccesary references. Some points just require one reference; I am unsure the use of all 114 refered works.

Author Response

Comments and Suggestions for Authors

The presentation of a GPR39 agonist as exerting effects related to OAB in female rats is attractive and clear. I suggest edition to improve relevance.

  1. a) Please highlight the possible mechanisms involved in the observed effects, albeit if not there were included on the measured variables. In this sense, I hope edition in introduction and discussion regarding potential indirect effects in Autonomous Nervous System and the GABA-Glu balance.

Understanding the pathogenesis of an overactive bladder (OAB) is essential for the development of targeted treatments. The proper functioning of the urinary tract, including urine storage and release, relies heavily on central nervous system (CNS) pathways. These pathways are modulated by excitatory and inhibitory neurotransmitters that regulate the micturition reflex [Yoshimura et al., 2014]. Disruptions in any phase of the micturition reflex, whether due to morphological changes in the nerves, smooth muscle, or urothelium, can contribute to OAB [Wróbel, 2015]. One mechanism implicated in detrusor overactivity is the loss of inhibitory control, which is potentially caused by impaired brain blood supply or increased excitatory signaling. Both the GABAergic and glutamatergic systems play significant roles. Glutamic acid, a key neurotransmitter in the micturition reflex, also serves as a precursor for the inhibitory neurotransmitter gamma-aminobutyric acid (GABA) [Kontani & Ueda, 2005]. Emerging evidence suggests that GABA analogs may ameliorate bladder dysfunction and that glutamine may prevent structural damage to the bladder. Additionally, the link between OAB and depression is well established with shared neurochemical underpinnings. Although the pathophysiology of depression remains complex, it likely involves imbalances in glutamate and GABA signaling, both of which are key targets of antidepressants [Fogaça et al., 2023]. The GPR39 receptor has garnered attention for its antidepressant-like effects that are mediated by zinc. Zinc antagonizes NMDA glutamate receptors, inhibits excitatory transmission, and contributes to the antidepressant effects [Skolnick et al.; Pilc et al., 2013]. Zinc deficiency has been linked to elevated glutamate levels, disrupting the GABA-glutamate balance and contributing to depressive symptoms [Takeda et al., 2008]. Importantly, GPR39 activation leads to the increased synthesis of 2-arachidonoylglycerol (2-AG), which inhibits glutamate release and restores this balance [Perez-Rosello et al., 2013]. Animals lacking GPR39 showed reduced 2-AG synthesis, exacerbating excitatory imbalance. Further supporting this mechanism, GPR39 activation upregulates potassium-chloride cotransporter (KCC2) expression in the hippocampal CA3 region, enhancing inhibitory GABAergic transmission by hyperpolarizing neurons through increased chloride influx [Chorin et al., 2011]. This mechanism is pivotal for mitigating excitotoxicity and enhancing neural plasticity, both of which are critical for antidepressant efficacy. In summary, the interplay between GPR39, zinc, and the GABA-glutamate system offers insight into OAB pathogenesis and its frequent co-occurrence with depression. By modulating glutamate release and enhancing GABAergic signaling, GPR39 may be a novel therapeutic target for these interconnected conditions.

  1. b) Consider the content of the following articles to edit and generate a deeper discussion in regards of the potential mechanisms related with the effects in bladder.

Fogaça, M. V., Wu, M., Li, C., Li, X. Y., Duman, R. S., & Picciotto, M. R. (2023). M1 acetylcholine receptors in somatostatin interneurons contribute to GABAergic and glutamatergic plasticity in the mPFC and antidepressant-like responses. Neuropsychopharmacology48(9), 1277-1287.

Yoshimura, N., Miyazato, M., Kitta, T., & Yoshikawa, S. (2014). Central nervous targets for the treatment of bladder dysfunction. Neurourology and Urodynamics33(1), 59-66.

Kontani, H., & Ueda, Y. (2005). A method for producing overactive bladder in the rat and investigation of the effects of GABAergic receptor agonists and glutamatergic receptor antagonists on the cystometrogram. The Journal of urology173(5), 1805-1811.

We thank the reviewer for highlighting the valuable references. We have reviewed the suggested articles and incorporated relevant insights into the discussion. Specifically, we expanded on the mechanisms underlying the effects observed in the bladder, particularly focusing on the roles of GABAergic and glutamatergic plasticity and the central nervous system pathways involved in micturition reflex regulation, as detailed in the recommended studies.

  1. c) Please add the meaning of the labels used in the charts. Albeit if it is predictable. Check the charts legends for the use of adequate asterisks meaning. For example, in Fig. 2, the *p=0.05 is in the legend, but no presented in the chart.

We thank the reviewer for pointing this out these important clarifications. We have carefully revised all the figure legends to ensure that the meanings of the labels are clearly explained. In particular, For Fig. 2, we corrected the figure legend to accurately reflect the data presented in the chart, ensuring alignment between the visual representation and its description. Specifically, we have deleted the *p = 0.05 significance level to the chart where applicable. Additionally, we reviewed the legends for all figures to ensure consistent usage of asterisks and clear explanations of their significance levels. These changes were intended to enhance the clarity and accuracy of the figures. We hope that these revisions address the reviewers’ concerns.

  1. d) Check the use of a the neccesary references. Some points just require one reference; I am unsure the use of all 114 refered works.

We thank the reviewer for bringing this to our attention. We thoroughly reviewed the manuscript to ensure that all references were used appropriately, and redundantly cited references were removed. The following adjustments were made. Introduction: References in this section have been carefully examined and redundant citations have been removed. Specifically, references 1–34 were reassessed and repeated citations were eliminated. These adjustments have been marked in green in the manuscript for clarity. Materials and Methods: We identified two missing references noted by the reviewer. These have now been added to the reference list as items 35 and 36, and are cited appropriately in the text. Discussion: References 41–114 have been carefully checked for relevance, redundancy, and necessity. Any repeated or unnecessary citations were removed to streamline the manuscript. We are confident that these corrections address the reviewers’ concerns, ensuring that all cited works are necessary, appropriately used, and relevant to the content of the manuscript.

Reviewer 2 Report

Comments and Suggestions for Authors

The authors present an interesting study in which the relationship between bladder activity and the mental state of depression is examined. Briefly, the authors employ an animal model in which corticosterone was administered daily in order to induce a ‘depression’ state in the animals and then enable examination of this state on molecules and behavioural traits associated with depression, bladder activity, and similar. In parallel, a group of animals was administered TC-G 1008 to examine the influence of a specific GPR39 agonist on the same indices, while a further group in receipt of both was established to determine whether a relationship existed between the two. Overall, the result indicate that molecules such as TC-G 1008 that activate GPR39 may have value in treating bladder conditions, in addition to treating conditions such as depression.

Overall I thought this was a well constructed manuscript with clear goals and a good study design. As such, my pints are minimal but I would like clarity on the following:

1.      The authors utilised female mice for this study – is there any particular reason this decision was made? It is good that the authors mentioned this as a limitation, but why were male rats not included for example, or why were females chosen over males?

2.      In terms of the concentrations of the TC-G 1008 compound administered and the dosing regime, were these amounts based on a previous study or how were these effective concentrations determined?

Author Response

Comments and Suggestions for Authors

The authors present an interesting study in which the relationship between bladder activity and the mental state of depression is examined. Briefly, the authors employ an animal model in which corticosterone was administered daily in order to induce a ‘depression’ state in the animals and then enable examination of this state on molecules and behavioural traits associated with depression, bladder activity, and similar. In parallel, a group of animals was administered TC-G 1008 to examine the influence of a specific GPR39 agonist on the same indices, while a further group in receipt of both was established to determine whether a relationship existed between the two. Overall, the result indicate that molecules such as TC-G 1008 that activate GPR39 may have value in treating bladder conditions, in addition to treating conditions such as depression.

Overall I thought this was a well constructed manuscript with clear goals and a good study design. As such, my pints are minimal but I would like clarity on the following:

  1. The authors utilised female mice for this study – is there any particular reason this decision was made? It is good that the authors mentioned this as a limitation, but why were male rats not included for example, or why were females chosen over males?

We thank the reviewer for the insightful comment. In our study, we chose to use female rats because of the significantly higher prevalence of overactive bladder (OAB) symptoms in women than in men. Studies have estimated that approximately 52% of adult women experience OAB symptoms, making this a critical population for study [Melotti et al., 2017]. Additionally, the co-occurrence of depression and OAB is particularly prevalent in women, aligning with the focus of our study. Male participants were excluded because the pathophysiology of OAB differs significantly between the sexes. In men, OAB is more frequently associated with bladder outlet obstruction (BOO) secondary to prostate enlargement, which introduces distinct etiological factors that are not directly relevant to our research model. The animal model used in this study is a well-established, replicated, and validated model from our previous studies. These factors, along with the focus on female-specific prevalence and mechanisms of OAB and its comorbidities, informed our decision to use female rats exclusively in this investigation.

  1. In terms of the concentrations of the TC-G 1008 compound administered and the dosing regime, were these amounts based on a previous study or how were these effective concentrations determined?

We appreciate the reviewer’s comment regarding the dosing regime and concentration of TC-G 1008 used in our study. In this study, TC-G 1008 (15 mg/kg; Tocris Bioscience, Great Britain) was administered intraperitoneally. Details regarding the source, dose, and route of administration are provided in the "Drugs used in the study" section under "Materials and Methods." The chosen dose of TC-G 1008 was based on previously published findings, particularly the study by Starowicz et al. (2023), which explored the role of the GPR39 zinc receptor in modulating glutamatergic and GABAergic neurotransmission [Starowicz, G., Siodłak, D., Nowak, G. et al. Pharmacol. Rep 75, 609–622 (2023); https://doi.org/10.1007/s43440-023-00478-0]. This dose has been shown to be effective in inducing antidepressant-like effects in behavioral tests such as the Forced Swim Test (FST). Additionally, we performed preliminary validation experiments to confirm the efficacy and relevance of the 15 mg/kg dose in the context of our study. This preliminary study confirmed that the chosen dose allowed us to accurately evaluate the antidepressant effects of the GPR39 agonist in our model. We hope that this clarifies the rationale for selecting this dosing regime and provides a detailed context for the concentrations used.